# Study on Rationality of Public Fitness Service Facilities in Beijing Based on GIS

**Xuefeng Tan** [†], **Chenggen Guo** [†] and **Pu Sun** *

School of Physical Education and Sports, Beijing Normal University, North Taipingzhuang Street, Beijing 100875, China
* Correspondence: sunpu@bnu.edu.cn
† These authors contributed equally to this work.

**Abstract:** Public fitness service facilities are one of the most important factors affecting national fitness. Analyzing the distribution characteristics and influencing factors of public fitness service facilities is of great significance to optimizing public fitness services. In this paper, using the population concentration index and consistency test, recent proximity index, coupling coordinated development model, buffer analysis, and correlation analysis, combined with ArcGis 10.7 software research, we found that: the site layout of public fitness service facilities in the main urban area of Beijing is relatively reasonable and has good coupling, and each area becomes a cluster trend; In the main urban area of Beijing, the public fitness service facilities and the transportation line are combined better; Public fitness service facilities in the main urban area of Beijing are well combined with schools (kindergartens, primary schools), with schools as the center; The public fitness service facilities in the main urban area of Beijing are well combined with medical sites, and some marginal areas are not covered; Public fitness service facilities in the main urban area in Beijing show uneven distribution when drawing with influential business districts; The layout of public fitness service facilities in the main urban area of Beijing is positively related to the housing price.

**Keywords:** GIS; residents; public fitness services; Beijing





## 1. Introduction

As people's participation in national fitness continues to increase, and diversified fitness needs continue to emerge, the issues of how to effectively reorganize and classify resources, how to better equip sports resources and how to protect the fitness needs of the general public also arise and need to be solved. China attaches great importance to the health of its people. China is committed to achieving the goal of a healthy China, and this goal cannot be achieved without national fitness [1]. National fitness is inseparable from the configuration of fitness facilities, and good public fitness facilities and their layout are the guarantee for national fitness. Public fitness facilities are social infrastructures that provide services for residents to participate in sports, and its layout and radiation range reflect the resource ratios and levels in public services to a certain extent [2]. Studying the current situation and trend of the layout of urban public fitness facilities can reflect, to a certain extent, whether public fitness services are fair and whether the public fitness needs of urban residents are met. It is also an important factor to evaluate the goodness of the whole urban public fitness service system. In the process of promoting the development of national fitness, it is necessary to build a perfect public service supply system as the basis, comprehensively optimize the efficiency of social resource coordination and allocation quality, continuously optimize the guarantee mechanism and service system in order to effectively enhance people's participation enthusiasm and physical quality, and lay a good foundation in order to comprehensively reach the goal of a healthy China [3].

GIS software has been widely used to investigate the location of public service facilities, the equity of health facilities, the location of optimal facilities for tourism, community

service facilities, and the design of public transportation systems [4–8], with good results. In sports-related research, GIS software is usually applied in the study of public sports space, sports space distribution, and sports industry [9–11]. Through literature collection, some scholars have used GIS software to study the accessibility of community public sports fields in the central city of Shanghai, China to evaluate the accessibility of existing community public sports fields in the central city of Shanghai, and to propose improvement strategies and solutions to refine the principles of community public sports field layouts in the main city of Shanghai [12]. Some scholars also analyzed the spatial layout of the sports industry in Wuhan based on the knowledge of geographic economics based on the GIS technology platform and proposed the feasibility [13]. In summary, it is found that the use of GIS software can provide a clear visual analysis of the sports industry as well as the layout of sports fields. However, there are fewer studies using GIS to analyze public fitness facilities for residents, and some studies have used GIS software as well as knowledge of population geography in analyzing the current situation of public sports service facilities in Chongqing, and have proposed feasible opinions for urban planning [14]. There are some limitations in the study, such as not making full use of the functions of GIS software, less visualization and analysis that can establish buffer zones, etc., for deeper profiling. Beijing, as the economic, political, and cultural center of China [3], has a pivotal position and demonstration effect in China. As a leader in responding to and implementing policies in China, Beijing's public fitness service system has strong research significance.

In summary, this paper is based on the population concentration index and consistency test commonly used in population geography to determine the layout of public fitness service facilities and the number of people and facilities in the main urban area of Beijing for a consistency test to determine whether it is reasonable, as well as the ratio of the observed distance of the nearest neighbor index to the reasonable expected distance. A visualization analysis was conducted using ArcGIS 10.7 software to analyze the public fitness service facilities in Beijing and to provide feasible opinions on the development of public construction services in the main city of Beijing.

## 2. Data Source and Study Methods

### 2.1. Data Source

This study obtained information from the website of the Beijing Public Health Service Platform, the official website of Beijing Municipal Education Bureau, the network map, and Beijing Municipal Bureau of Statistics. Google Earth converted the research objects into latitude and longitude and into GIS to obtain the required geographical layer, and the accuracy of the geographical layer was improved by comparing the Beijing administrative map.

The total population of the main urban area of Beijing is 11.236 million, with a total administrative area of 1385 square kilometers, and the number of residential public fitness venues is 3529. The specific regional distribution is shown in Table 1.

**Table 1.** Basic statistics of all urban areas in Beijing.

| Area | National Fitness Project (Film) | Open Fitness Venues (Individual) | Special Activity Site (Film) | Trail (Piece) | Community Fitness Club (Individual) | Total Number of Public Fitness Facilities (Tablets) | Total District Administrative Area (km²) | Total District Number (10,000) | Population Density (Human/km²) |
|---|---|---|---|---|---|---|---|---|---|
| Dongcheng District | 147 | 13 | 10 | 4 | 7 | 181 | 42 | 79.4 | 18,904.76 |
| Xicheng District | 320 | 0 | 15 | 7 | 9 | 351 | 51 | 113.7 | 222,941.1 |
| Chaoyang District | 869 | 0 | 103 | 10 | 12 | 994 | 471 | 347.3 | 7373.67 |
| Haidian District | 948 | 0 | 230 | 18 | 14 | 1210 | 431 | 323.7 | 749.42 |
| Fengtai District | 454 | 0 | 111 | 6 | 8 | 579 | 304 | 202.5 | 6661.18 |
| Shijingshan | 210 | 0 | 0 | 3 | 1 | 214 | 86 | 57 | 6627.91 |

Study Area

As shown in the Figure 1, the area selected in this study is the main urban area of Beijing, which includes Dongcheng District, Xicheng District, Chaoyang District, Haidian District, Fengtai District, and Shijingshan District, respectively.

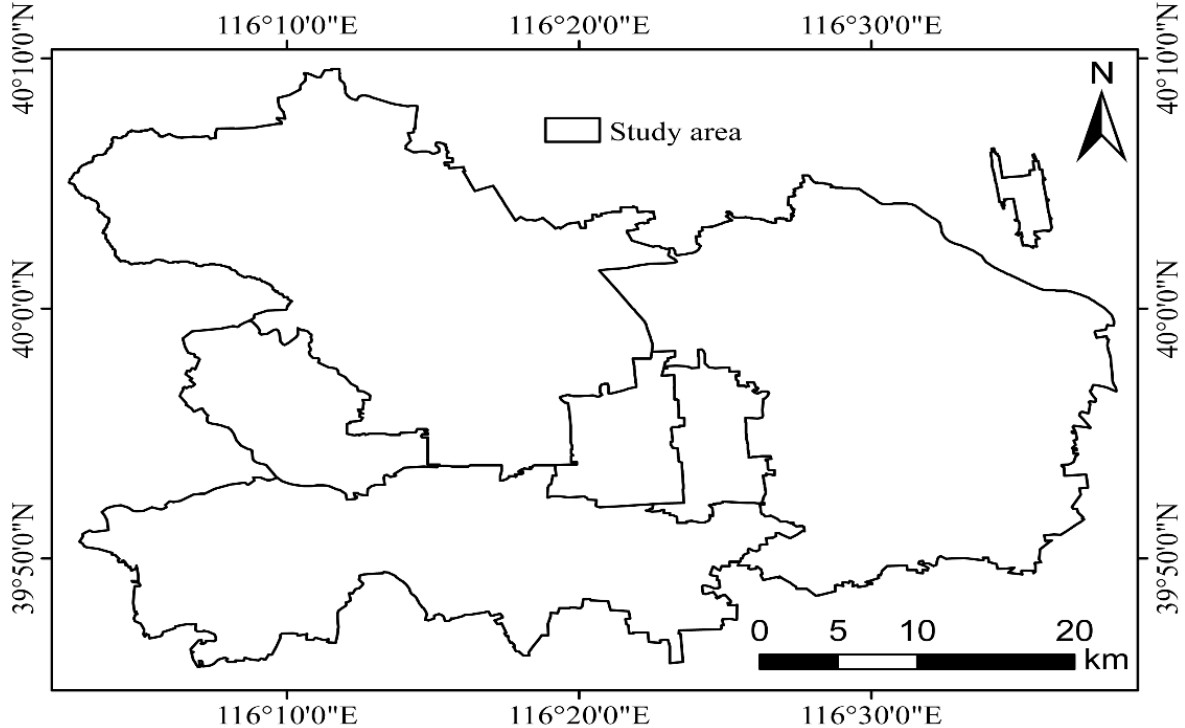

**Figure 1.** Major urban areas in Beijing.

*2.2. Literature Review Method and Article Research Process*

Through the retrieval network, the articles on public construction services published in the core journals were sorted out to lay the foundation for this research. The flow chart of the specific article research methodology is shown in Figure 2.

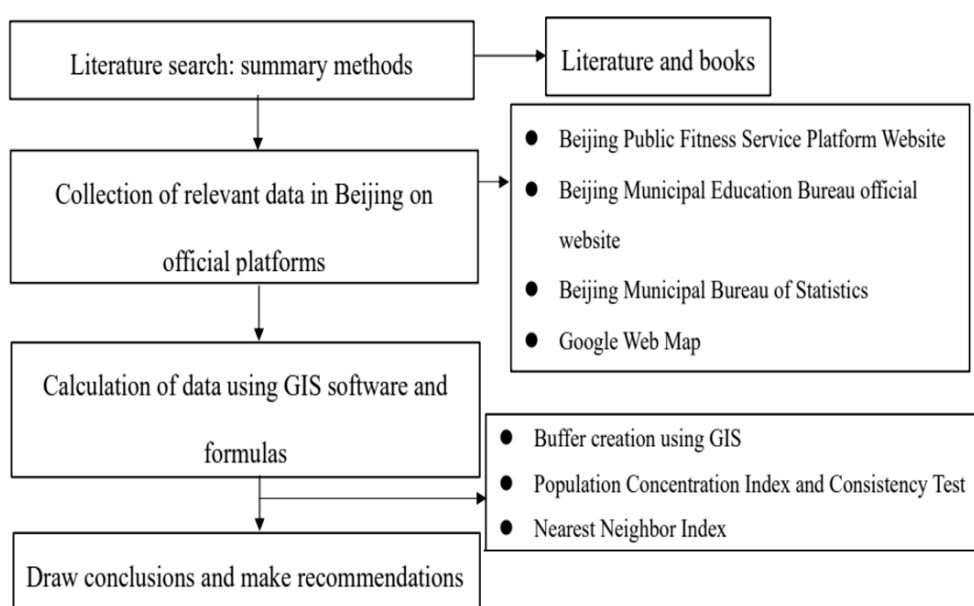

**Figure 2.** Research Flow Chart.

### 2.2.1. GIS Spatial Analysis

ArcGIS is a common software for GIS processing [15], Spatial feature analysis was performed by coordinate mapping of the statistical target information into GIS software. The public fitness service facilities counted in this article refer to the national or community investment completed and put into use, and the audience object is all residents. The simple profit-making stadium will not be included in these statistics.

### 2.2.2. Population Concentration Index and Consistency Test

To determine whether the layout of public fitness service facilities in an area is reasonable, the size of the population (population concentration index) and the number of facilities will be considered [16]. The consistency test is a common means of analyzing geographical space in geography, and is usually used in homogeneous studies of regional population and regional areas [14].

The formula of population concentration is:

$$\Delta p = \frac{1}{2} \sum_{i=1}^{N} \left| \frac{P_i}{\sum P_i} - \frac{S_i}{\sum S_i} \right| \times 100\% \tag{1}$$

$P_i S_i \sum P_i \sum S_i$: among them is the respective regional area, the total number of representatives, and the total amount of land owned. (1) When $\Delta p$ approaches 0, the population and land reach a harmonious state. (2) When $\Delta p$ is close to 1, it indicates that the state of population and land is unbalanced and people are in a concentration trend.

According to the formula, the identity index of public fitness service facilities and population distribution is:

$$R_{xy} = \left[ 1 - \frac{1}{2} \sum_{i=1}^{N} \left| \frac{X_i}{\sum X_i} - \frac{Y_i}{\sum Y_i} \right| \right] \times 100\% \tag{2}$$

$X_i Y_i \sum X_i \sum Y_i$: among them, $R_{xy}$ represents the index of identity, expressed as the number of districts, and is the number of public fitness services in the corresponding region, the total number of people as the study object, and the total number of public fitness services. Therefore, when the number is closer to 100, public fitness services are more concentrated.

### 2.2.3. Nearest Proximity Index

The nearest proximity index is the ratio representing the average observed distance to a reasonably expected distance [17]; exponential $R > 1$, the larger index is discrete distribution, $R < 1$, the smaller exponent is aggregation distribution, $R = 1$ is a random distribution [18].

Expression formula is:

$$R = \frac{r_1}{r_2} = 2\sqrt{D} \tag{3}$$

Among these, $R$ is expressed as the nearest neighbor index, $r_1$ represents the actual nearest neighbor point distance, $r_2$ represents the theoretical closest point distance, and $D$ is the point density.

### 2.2.4. Mathematical Statistics and Analysis

According to the information of the Beijing public fitness service platform, the layout and distribution of public construction service facilities in the main urban area were studied with GIS software as the main analysis tool. Part of the data were analyzed for statistical analysis using SPSS software. Statistical results and analysis results are presented graphically.

2.2.5. Buffer Analysis Method

Buffer analysis is commonly used in GIS systems to geographically describe the impact of targets on the radiation range, and is an important way to solve the problem of spatial proximity. In this study, the rationality of the layout of public fitness service facilities in the main urban area of Beijing was explored by taking medical treatment centers, schools, and business districts with the buffer radius of 2 km, 1 km, and 2 km, respectively.

## 3. Results of the Study

### 3.1. General Layout of Public Fitness Service Facilities in the Main Urban Area of Beijing

In general, Haidian District has the largest total number of public fitness facilities, and Dongcheng District has the least. The total numbers in Dongcheng District and Shijingshan District are relatively small, and the numbers in Haidian District and Chaoyang District are relatively large. Second, if an idealized spatial distribution is desired, the connection between morphological distribution and population density is important. Under normal circumstances, population density, number of sites, and administrative area should be positive and linear: as the number of people increases, the number of public fitness facilities should also be large; if the number is small, the number of public fitness facilities will decrease. According to Table 1, the size of the population, administrative areas, and public fitness facilities in the main urban areas of Beijing are positively correlated. However, this is a macrostatistic, and details regarding the block and the community should be further analyzed.

As can be seen from the official division of the Beijing public fitness service platform, the public fitness service facilities can be roughly divided into national fitness projects, open fitness venues (non-simple profit), special activity venues, trails, and community fitness clubs. It can be found from Table 1 that Dongcheng District has the smallest number, but has the most open sports venues. No other main urban areas, especially Haidian District and Chaoyang District with their large numbers, lack only open fitness venues with the large number of people and the number of public fitness service facilities.

### 3.2. Analysis of the Spatial Layout of Community Sports Public Facilities in the Main Urban Area of Beijing

3.2.1. The Spatial Layout of Community Sports Public Facilities in the Main Urban Area of Beijing City Is Reasonable

Rationality of distribution in geography often uses a coupling test and consistency analysis, which is also the main reference in this paper. Equation (1) only uses the formula to calculate the population concentration index of the main urban area of Beijing, and derive = 0.10471 in the main urban area of Beijing. It is known to be reasonable near 0 and unreasonable at 1. It shows that the population and land area in the main urban area are coupled. Equation (2) uses the formula calculate the combination degree of population and public fitness service facilities in the main urban area of Beijing, and obtained $Rxy$ = 93.53. It is known that when approaching 100, it shows that the population is better combined with public fitness service facilities. Therefore, the population in the main urban area of Beijing is well coupled with Beijing public fitness service facilities, which belongs to a more reasonable form.

$$\Delta P = \frac{1}{2} \sum_{i=1}^{N} \left| \frac{P_i}{\sum P_i} - \frac{S_i}{\sum S_i} \right| \times 100\% \Delta P \Delta P R_{xy} = \left[ 1 - \frac{1}{2} \sum_{i=1}^{N} \left| \frac{X_i}{\sum X_i} - \frac{Y_i}{\sum Y_i} \right| \right] \times 100\% R_{xy} R_{xy} \quad (4)$$

In order to further verify the rationality of the layout of public fitness service facilities in Beijing, the data of the public fitness service facilities is converted into GIS in Figures 3 and 4. According to the distribution diagram and scatter map, and with the density, the distribution point is sparse and increases, which can infer economic development level and the political–geopolitical relationship are also the factors affecting the distribution of public construction service facilities.

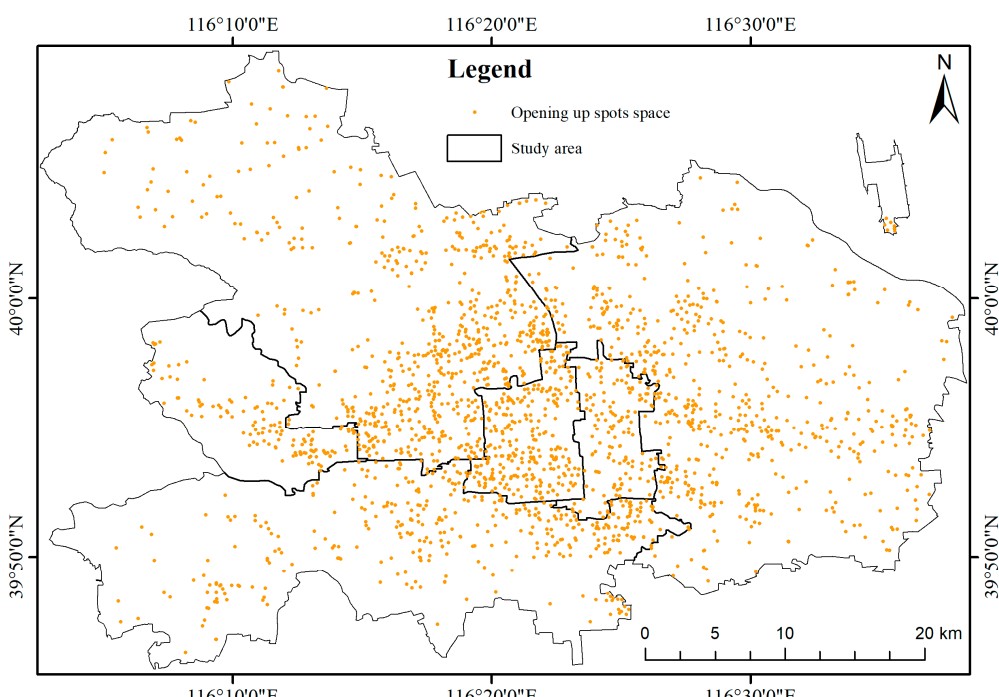

**Figure 3.** Distribution diagram of Figure.

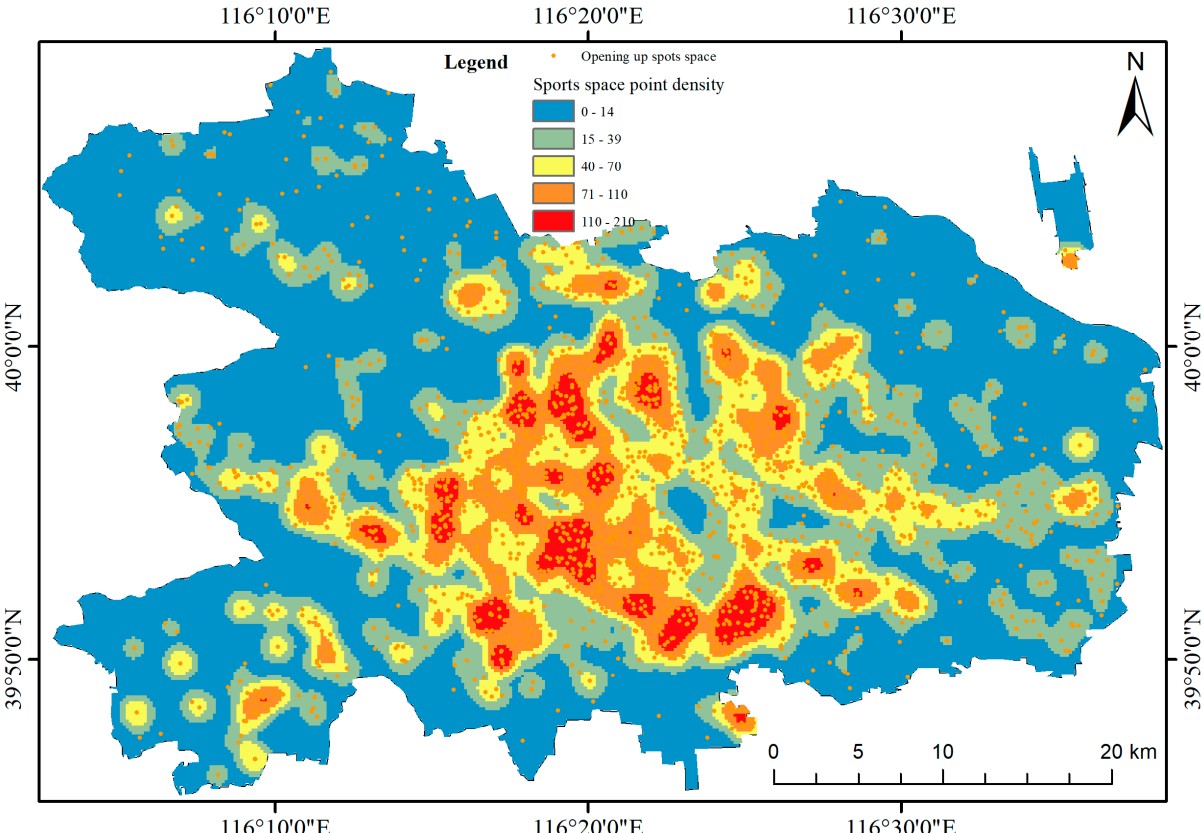

**Figure 4.** Hot spot analysis.

To verify the correctness of the GIS analysis plots, the correlation and regression analysis was used. Results are shown in Tables 2 and 3.

**Table 2.** Correlation coefficient analysis.

| Title 1 | Total Fitness Facilities | Total Number |
|---|---|---|
| Total fitness facilities | 1 | |
| Total number | 0.9702 | 1 |

**Table 3.** Results of the regression analysis.

| Variable | Estimate Coefficient | Standard Error | *t* Statistics | *p* |
|---|---|---|---|---|
| Total number | 3.310097 | 0.413543 | 8.004234 | 0.0013 |
| C | −31.70414 | 90.74209 | −0.349387 | 0.7444 |
| R square | | 0.941235 | | |
| F statistics | | 64.06776 | | |
| P (F statistics) | | 0.001321 | | |

The correlation analysis of the total number of the total fitness facilities in six districts, and the six districts in the main urban area of Beijing, are shown in Table 2, which shows the correlation coefficient of the total number, 0.9702, which is greater than 0. This shows that there is a positive relationship between the total number of fitness facilities and the total number of people. Next, the regression analysis method was continued to conduct the in-depth analysis of the relationship between the total number of fitness facilities and the total number of people.

As shown in Table 3, regression analysis using EVIEWS software, with the total number of fitness facilities as the independent variable, showed that the R party estimated by the model reached 0.941235 with a high goodness of fit. The F statistic value of 64.06776 and the corresponding *p* value of 0.001321, less than 0.05, indicated that the linear relationship between the total number of fitness facilities and the total number is significant. In further analysis, the estimated coefficient of the total number is 3.310097, and the *p* value of the significance test is 0.0013, which is less than 0.05, indicating that the total number has a significant positive impact on the total number of fitness facilities. Therefore, the higher the total number of people in the main urban area of Beijing, the higher the total number of fitness facilities.

As calculated from the nearest proximity index: Haidian District: 0.281742; Chaoyang District: 0.366364; Dongcheng District: 0.578833; Xicheng District: 0.541042; Fengtai District: 0.268452; Shijingshan District: 0.298717. Their R values are less than 1, indicating a cluster distribution in each region, but the relatively uniform distribution is calculated from the six regions.

In conclusion, the results of the coupling relationship test and consistency test are basically consistent with those of GIS distribution map. It shows that the distribution of population and public construction service facilities in the main urban area of Beijing is more reasonable. However, whether the distribution of each block, street, and community is reasonable remains uninvestigated.

3.2.2. Research on Site Layout of Public Fitness Facilities and Transportation Lines in the Main Urban Area of Beijing

Transportation is an important factor for residents to choose public fitness venues and facilities. Public fitness service facilities are positively related to the degree of transportation convenience, which is one of the important reference factors affecting the layout of public fitness service facilities. As shown in Figure 5, this paper draws the Beijing road network and the public fitness service facilities in the main urban area of Beijing through GIS. As can be seen from the figure, the public fitness service facilities in the main urban area of Beijing are mostly around the Beijing road network, and in 1 km of the urban traffic trunk as the buffer area are approximately 68% of the public fitness service facilities in the buffer area. Studies show that the traffic dependence of public fitness service facilities in the main

urban area of Beijing is ranked as: sports stadium > community national fitness project [19]. In this study, sports venues, fitness clubs, and open special activity venues have the same dependence on traffic, from which the stadiums, fitness clubs, open special activity venues > trail > national fitness project was launched. In general, the public fitness service facilities in the main urban area of Beijing rely on the traffic line, and can cover each fitness point to meet the travel needs of residents.

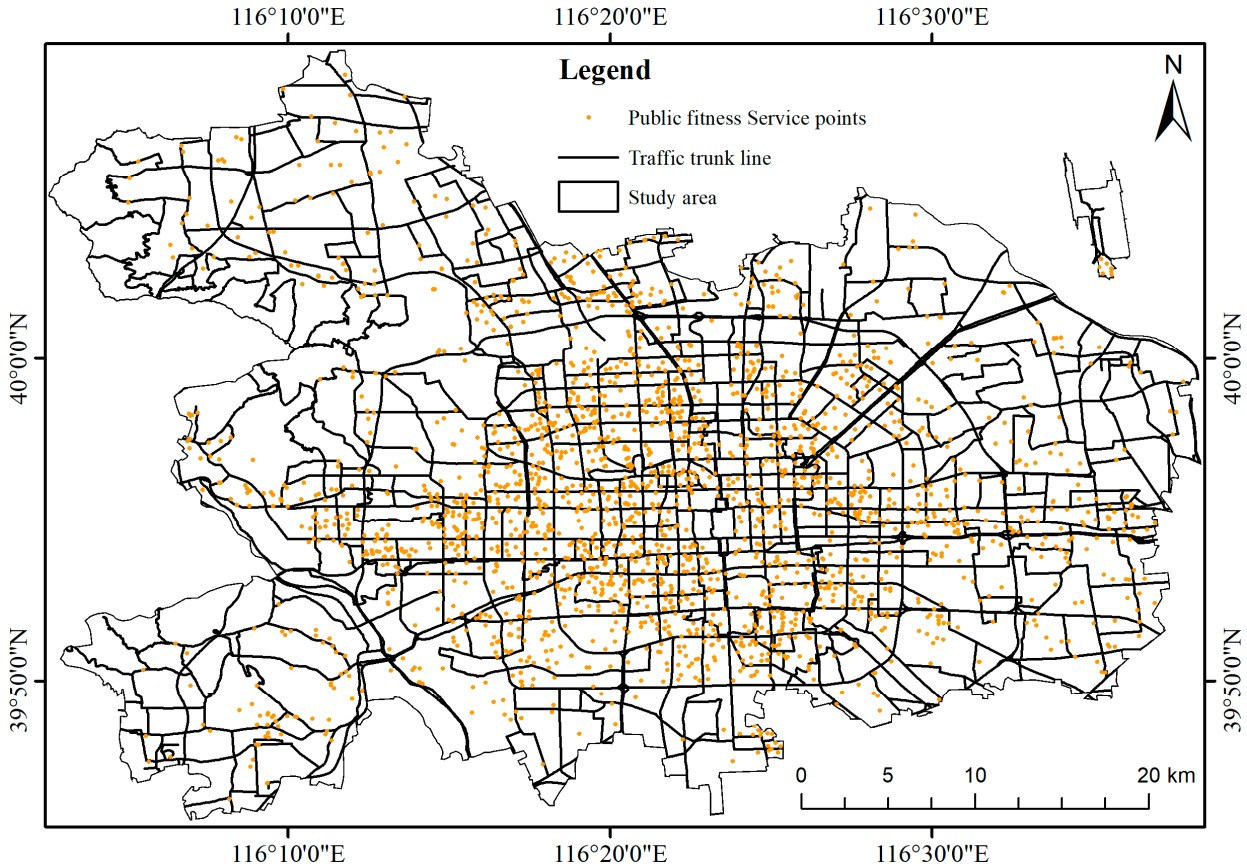

**Figure 5.** Beijing public fitness service facility points and transportation map.

### 3.2.3. Research on the Relationship between Site Layout of Public Fitness Facilities and Large Medical Sites in the Main Urban Area of Beijing

Medical care and public fitness are a mutually reinforcing and inseparable organic whole [20]. Promoting a good synergy between public fitness services and medical services is the key to address the health of the whole population at the grassroots level [21]. Medical treatment is the backbone of residents' peace of mind about fitness, including rehabilitation physiotherapy services, rehabilitation knowledge promotion, and physical fitness testing. In addition, health care and public fitness are a kind of integration. The Integration of fitness and health services is called "sports-medical integration", which enables the sports industry and the health industry to integrate, complement, and penetrate each other's superior resources so that they can jointly produce health-promoting effects. The integration of sports and health care is an important means to solve the problem of "health for all" and "fitness for all" [22]. As shown in Figure 6, this study counted the 1-A to 3-A hospitals in the main urban area of Beijing, and a buffer zone was created with a radius of 2 km from the hospital as the center. From the figure, it can be seen that 1- to 3-A hospitals with more complete medical resources and more complete departments can cover all fitness area points, showing a trend that spreads outward from the center, and there are uncovered areas at the edge of the main urban area. Medical treatment is, to a certain extent, the guarantee of public fitness; if there is a sudden accident, you can get medical treatment in a

relatively short time. The coverage of medical points is an effective means of transferring scientific exercise knowledge, first aid knowledge, and nutrition knowledge to residents. However, there are still some problems, such as the lack of medical point coverage in the periphery of the main city.

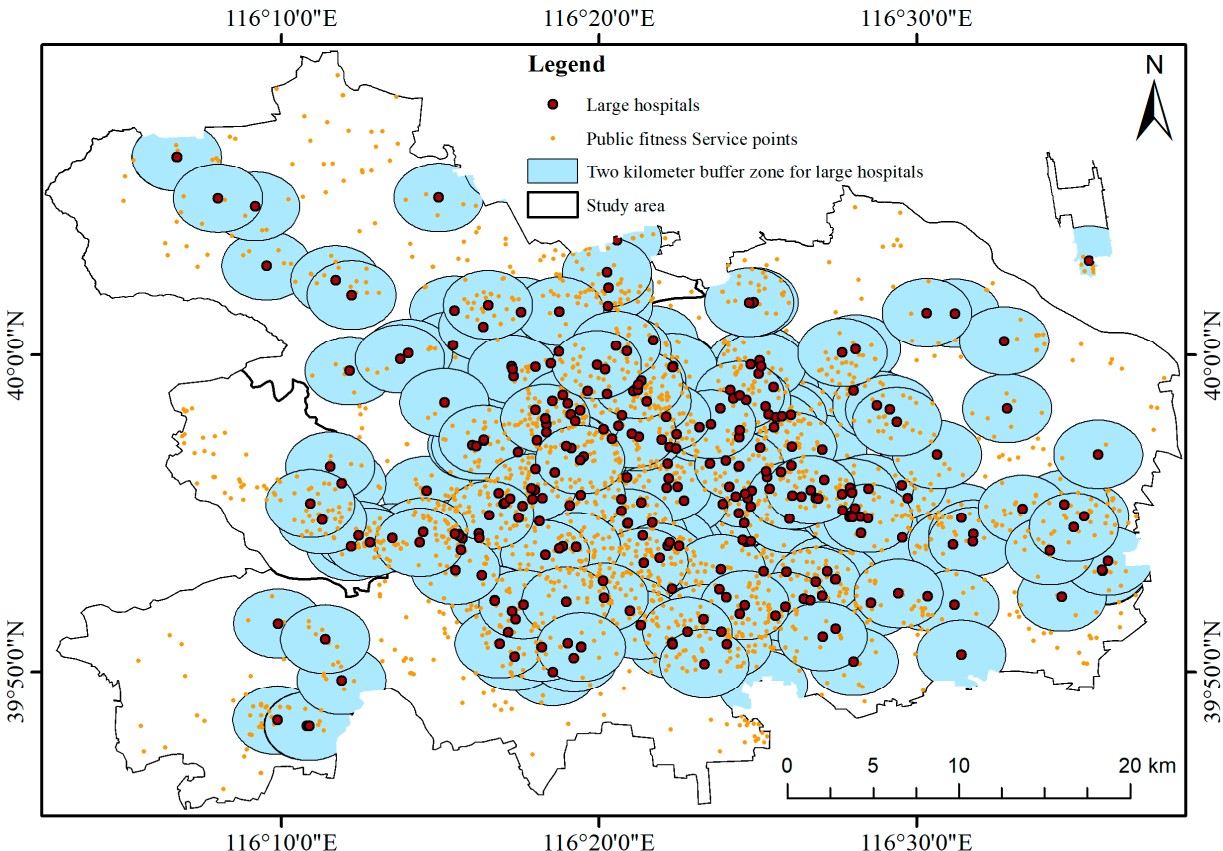

**Figure 6.** Public fitness service facilities and medical points in Beijing.

3.2.4. Relationship between Public Fitness Facilities Layout and Schools in the Main Urban Area of Beijing

As shown in Figure 7, this study took kindergartens and primary schools in the main urban area of Beijing as the central point, and a buffer map was made with the radius of 1 km. The reasons why kindergarten and primary schools were selected as the central points in this study are: (1) Kindergarten and primary school students are too young and need to be picked up from school to school. Therefore, can parents can exercise during the waiting period. (2) School health education and physical education are one of the most important means to achieve "national health" and "national fitness" [23], and kindergarten and primary school stages belong to the golden period of cultivating sports hobbies. If you can grasp this period well, it can lay a foundation for students' lifelong interest in sports. In addition, through physical education and health education in school, kindergarten and primary school students learn early, and whether there is a place to practice after school is also a question worth thinking about.

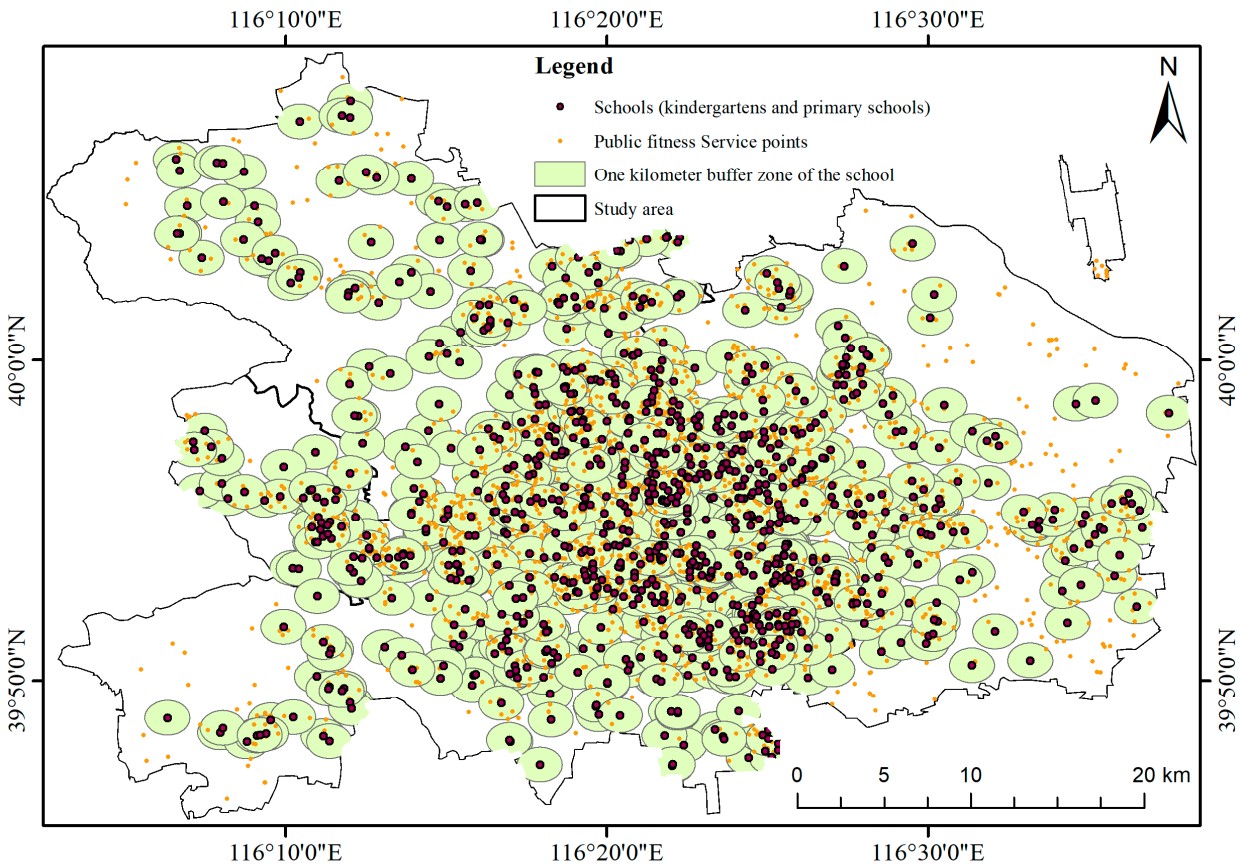

**Figure 7.** Relationship map between public fitness service facilities and schools in Beijing.

The figure shows that the overall coverage is high, with only a few marginal areas not covered. The closer it is to the Dongcheng District, the more densely the Xicheng District covers, showing a trend of spreading from the center to the outside. This phenomenon is positively related to the size of the area, population, and economic strength of the administrative region. In general, the layout of public fitness facilities in the main urban area of Beijing has a good relationship with schools.

3.2.5. Research on the Relationship between Public Fitness Facility Site Layouts and Business District and Housing Price in the Main Urban Area of Beijing City

Studies have shown that the combination of business circle and national fitness can form a good market environment [24]. The premium effect of public service facilities on the housing market is a key determinant of housing prices [25]. Studies have shown that the combination of business circle and national fitness is also conducive to the formation of the sports business circle [26]. The importance of the business district in a region can be drawn to public fitness. Studies have shown that the regional housing price has a certain impact on the spatial layout of leisure sports [19]. In summary, this paper analyzes the economic business districts and housing prices to draw the connection between public fitness service facilities and business district housing prices. As shown in Figures 8 and 9, this study takes the 10 influential business districts as the central points and the radius of 2 km as the buffer zone in addition to the regional housing price. As can be seen from Figure 8, these 10 influential business districts are relatively concentrated, have a concentrated trend, do not form a balanced distribution, and the coverage rate is not high. This phenomenon has a certain connection with politics, economy, and culture. As can be seen from Figure 9, the higher the housing price, the higher the number of public fitness service facilities, which may be in line with the improvement of supporting facilities. Combined with the

housing price, regional area, and population number, this distribution is in line with the real situation.

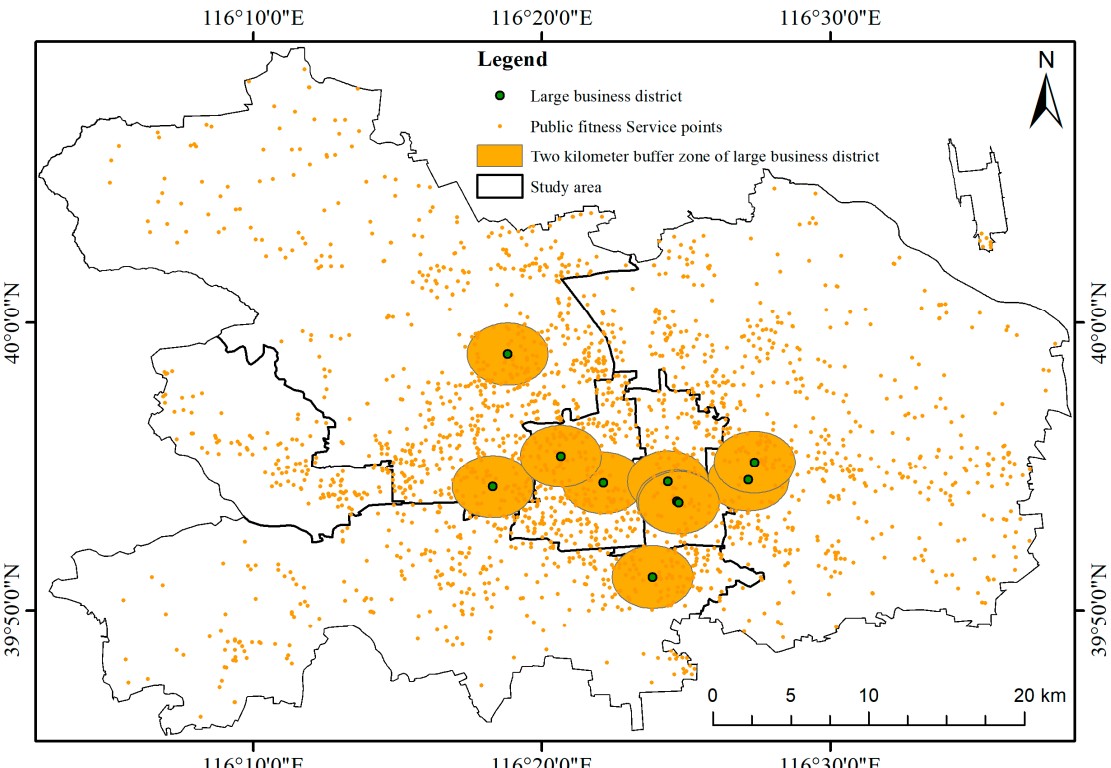

**Figure 8.** Relationship diagram between public fitness service facilities and the business circle.

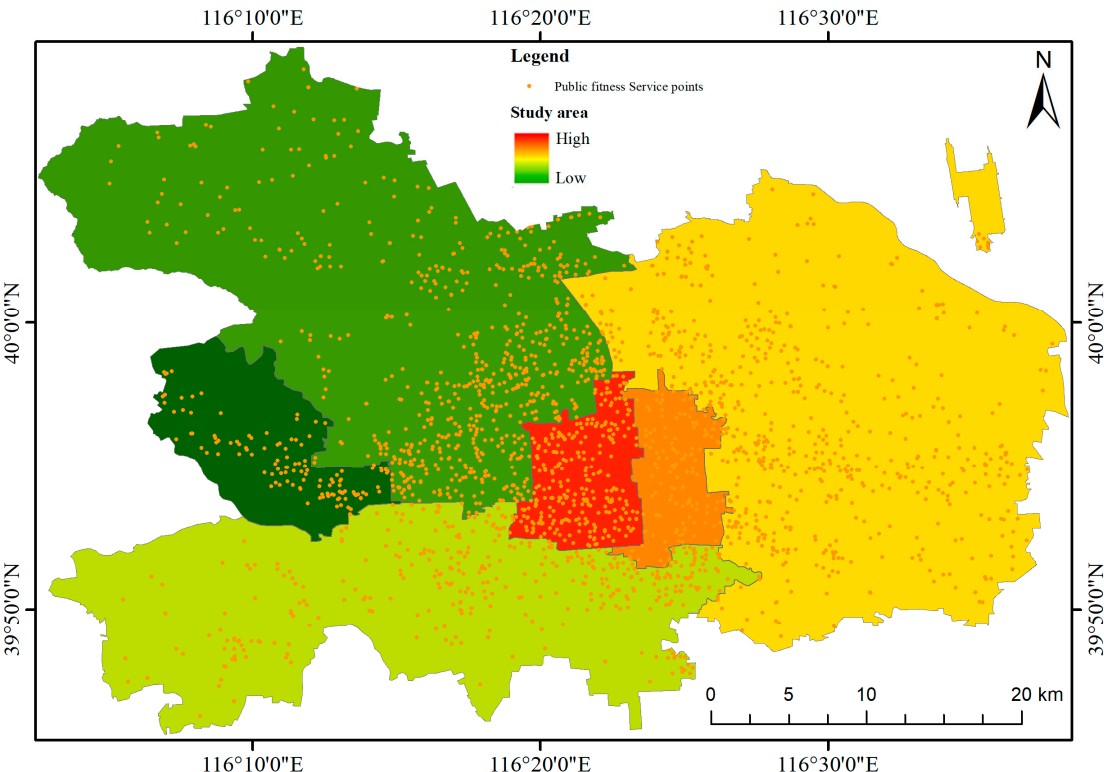

**Figure 9.** Relationship diagram between public fitness service facilities and district housing prices.

In summary, there are the above characteristics in economic aspects: (1) In the center of a large business circle, the distribution is uneven, and this phenomenon is normal to a certain extent. (2) From the perspective of housing price, regional area, and population number, the distribution and number of public fitness service facilities are relatively reasonable.

## 4. Discussion

### 4.1. Discussion of the Site Layout of Public Fitness Facilities in the Main Urban Area of Beijing

After the inspection of four inspection methods, the overall distribution of public fitness service facilities in the main urban area of Beijing is relatively reasonable, with good coupling and a positive correlation between the population density and the ratio of population-occupied sites in Beijing. However, during calculation analysis using the nearest proximity index, it was calculated that: Haidian District: 0.281742; Chaoyang District: 0.366364; Dongcheng District: 0.578833; Xicheng District: 0.541042; Fengtai District: 0.268452; Shijingshan District: 0.298717. Comparing the value with the reference value 1, we found that although the main urban area of Beijing is relatively average, it shows a concentrated trend after refining it to each area. The refinement of the distribution is not ideal. Secondly, the details found that there are obviously more public fitness service facilities in the Xicheng District, Dongcheng District, Haidian District, Chaoyang District, thus leading to an unbalanced image.

Therefore, the construction of public fitness service facilities should be adjusted, and there should be a uniform distribution of public construction service facilities to meet the requirements of different locations where residents exercise. Second, the main city area in the development should help each other, and there should be a reasonable allocation of resources for the coordinated development of each area, to the maximum extent. Finally, there are also significant issues in the types of public fitness service facilities. As shown in Table 4, there are too few open fitness venues and clubs, resulting in "fake prosperity" under the balance of distribution. The main urban area of Beijing should strengthen the construction of open fitness venues and community clubs in order to achieve balanced development, and to meet the fitness needs of different people as far as possible.

**Table 4.** Beijing Fitness Venue and Community Club.

| Area | Open Fitness Venues (Individual) | Special Activity Site (Film) | Community Health Club (All) | Fitness Clubs with National Words (All) |
|---|---|---|---|---|
| Dongcheng District | 13 | 10 | 7 | 1 |
| Xicheng District | 0 | 15 | 9 | 3 |
| Chaoyang District | 0 | 103 | 12 | 4 |
| Haidian District | 0 | 230 | 14 | 3 |
| Fengtai District | 0 | 111 | 8 | 1 |
| Shijingshan | 0 | 0 | 1 | 0 |

### 4.2. Discussion between the Layout of Public Fitness Facilities and Transportation Lines in the Main Urban Area of Beijing

It can be seen from the GIS diagram that the combination of transportation and public fitness service facilities in the main urban area of Beijing is good, which also shows that the traffic accessibility in the main urban area of Beijing is good. The reasons for forming such a situation are as follows: (1) The national fitness project accounts for a relatively large proportion of public fitness service facilities, which is built by the community as a unit. It can be concluded that it relies on the traffic planning of the municipal government in the main urban area of Beijing. (2) Open fitness venues, special activity venues, clubs and trails, convenient transportation, and whether there are supporting parking lots or nearby subway stations around are still worth considering. This not only affects the site selection of public fitness facilities, but also is one of the important factors affecting residents in participating in fitness.

In summary, after deliberation, it is concluded that the construction of supporting facilities needs to be fully considered in the site selection and construction of open fitness venues, special activity venues, clubs, and trails. There should be at least supporting parking lots, subway stations, and bus stations within a radius of these facilities as the center point. In order to make the residents exercise more convenient, we must improve the residents' desire to exercise.

### 4.3. Discussion of the Layout of Public Fitness Facilities and Large Medical Points in the Main Urban Areas of Beijing

The lack of medical point coverage on the edge of the main urban area is the main problem seen by GIS. If we cannot improve this in a short time, we can use other ways to help solve the problem. The main advantages of medical points and public fitness service facilities are to get the fastest and most effective treatment at the time of injury, and to guide residents to develop good health habits, fitness knowledge, and nutrition knowledge through the hospital health knowledge publicity. In summary, auxiliary help can be obtained through the following ways: (1) Due to the large number of community national fitness projects, we can organize community medical institutions and clinics to carry out irregular fitness knowledge and nutrition knowledge publicity. (2) Improving the reception capacity of community hospitals and clinics, especially the reception and pretreatment ability of sports injuries. (3) Expanding the number and quality of social sports instructors, give full play to the role of social sports instructors, and teach the correct exercise methods and intensity prediction in public fitness guidance, thus reducing the risk of sports injury, which is also conducive to the formation of fitness hobbies and habits. As shown in Table 5, the number of social sports instructors in the main urban area of Beijing is small, the grade distribution is uneven, and the overlapping and incomplete guidance projects need to be improved.

**Table 5.** Statistical table of social sports instructors in the main urban area of Beijing.

| Area | Fitness Enthusiasts | Level 3 Instructor | Second-Level Instructor | First-Level Instructor | National Instructor | Star Instructor | Total Number |
|------|---------------------|--------------------|-------------------------|------------------------|---------------------|-----------------|--------------|
| Dongcheng District | 4 | 1 | 3 | 6 | 0 | 4 | 18 |
| Xicheng District | 3 | 6 | 0 | 6 | 0 | 4 | 19 |
| Chaoyang District | 4 | 0 | 14 | 0 | 0 | 9 | 18 |
| Haidian District | 2 | 5 | 1 | 6 | 1 | 5 | 20 |
| Fengtai District | 0 | 6 | 4 | 4 | 0 | 3 | 17 |
| Shijingshan | 2 | 5 | 2 | 8 | 0 | 1 | 18 |

### 4.4. Discussion of the Layout of Public Fitness Facilities in the Main Urban Area of Beijing

The layout of public fitness service facilities in the main urban area of Beijing is well combined with the schools (kindergarten, primary school), and the layout is more reasonable. Only a few administrative border areas were not covered, but overall, the coverage was higher. This helps parents to get exercise before picking up their children and allows them to exercise from school.

When discussing the public fitness facility venues, the fitness venues and facilities in the schools are usually ignored. Some studies reported that we should pay attention to the radiation range of the school as the center and make reasonable use of the school fitness facilities as a way to supplement the public fitness service facilities. Beijing's main city is very rich in school resources, and they usually have complete sports venues and fitness facilities. This is the best choice for social public fitness service facilities site, especially in the center of the school radiation area, which can get high quality fitness choices and supplement the number of open venues. After universal vaccination, the school sports facilities can be orderly opened as planned to supplement the public fitness services.

### 4.5. Discussion of the Layout of Public Fitness Facilities and Business District and Housing Prices in the Main Urban Area of Beijing

The 10 business districts selected as influential in this study did not find a high coverage rate and had an uneven distribution. According to the GIS diagram of housing prices, the number of public fitness facilities was found to be positively related, and the distribution is more reasonable from an economic point of view. Studies show that the business circle can be the center of public construction service facilities for the infrastructure construction of "urban sports park" [27]. This model can be considered for reference in the main urban area of Beijing. An uneven distribution is found around the business circle, but if the scope is narrowed and the shopping mall serves as the center point, a uniform distribution may occur. The shopping mall as the center can create "a new shopping mall fitness mode", and some research has found that the shopping mall makes use of its own advantages and makes use of the shopping mall layout and walking combination to create shopping mall walking [28]. This is attracts residents to the shopping mall, but also achieves the purpose of fitness. Outdoor sports are essential for maintaining people's physical and mental health [29]. At the same time, the construction of public fitness service facilities around the mall makes residents need to choose between fitness and shopping, but can also allow them to choose both.

### 5. Advantage and Deficiencies

Advantages:

(1) This paper uses the population concentration index and consistency test, recent proximity index, coupled coordinated development model, buffer analysis, and correlation analysis, combined with the site layout of the main urban area in Beijing and the relationship between schools, transportation, medical treatment, and housing prices.

(2) The regions selected in this study have no research, and the subject knowledge of public construction service facilities on the layout of GIS and geography is less researched.

(3) The sample size of this data was large and the authoritative platform was adopted. Deficiencies:

(1) This study was generally biased towards the macro level, and did not significantly refine each district or even the street. The macroscopic results may be inconsistent with those after refinement.

(2) The scope of research is the main urban area of Beijing, and did not involve Daxing District, Tongzhou District, and Changping District.

(3) This study only considers the relationship between public fitness service facilities in the main urban area of Beijing and schools, medical treatment, transportation, business districts, and housing prices, and more factors can be considered in the later stage. (4) It was found in this study that the layout of public construction service facilities in the main urban area of Beijing is relatively perfect, but the utilization rate is still low according to the actual survey; we must track this problem in the future.

### 6. Conclusions

Conclusion: The size of the population and public fitness service facilities in the main urban area of Beijing are coupled. According to the GIS and statistical analysis, the spatial distribution of public fitness service facilities in the main urban area of Beijing is reasonable. There is an imbalance in the types of public fitness services, and the number of open venues is too small. Some of the public fitness facilities in each district have different grades and are of different quality. There is a problem in that the number of social instructors is small, the distribution is not proportional in each district, the service items are few, and the quality of the grade is poor. In summary, the analysis and countermeasures have put forward the corresponding countermeasures.

Outlook: The theory and tools of geography can be well analyzed to find out the rationality of the public fitness service system in the main urban area of Beijing. In this paper, the theme is set in the main urban area of Beijing to combine the size of the population

and the number of public fitness service facilities into a coupling relationship, but it is not known whether the depth to the streets and communities are also in a coupling relationship. Secondly, it is not known whether the spatial distributions of public fitness facilities in other districts, excluding the main urban area of Beijing, are still reasonable due to economic, demographic, and policy influences. In summary, each street and neighborhood can be detailed in future studies. The entirety of Beijing can be studied with the knowledge of geography if the data are sufficient, and the needs of each age group can be studied in detail.

**Author Contributions:** Conceptualization, P.S.; methodology, X.T.; software, X.T.; validation, X.T. and C.G.; formal analysis, X.T.; investigation, X.T. and C.G.; resources, X.T.; data curation, X.T. and C.G.; writing—original draft preparation, X.T.; writing—review and editing, X.T., C.G. and P.S.; visualization, X.T., C.G. and P.S.; supervision, P.S.; project administration, P.S.; funding acquisition, P.S. All authors have read and agreed to the published version of the manuscript.

**Funding:** This research was funded by 2020 Beijing Social Science Fund Planning Project: Research on the Content Construction of Beijing Community Residents' Fitness Service System from the perspective of deep integration grant number 20YTB012.

**Institutional Review Board Statement:** Not applicable.

**Informed Consent Statement:** Not applicable.

**Data Availability Statement:** The datasets analyzed during the current study are available from the corresponding author upon reasonable request.

**Conflicts of Interest:** The authors declare no conflict of interest.

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
