# Peer review of "Study on Rationality of Public Fitness Service Facilities in Beijing Based on GIS"

_sustainability, doi:10.3390/su15021496_

Round 1
Reviewer 1 Report
The introduction is short, inchoate and not very explanatory. It is necessary to increase the number of bibliographic references (mainly the most current ones) and I suggest revising the writing so that the text is more scientific. There is not a minimal literature review that allows the reader to know more about the history and importance of the subject. It remains to detail the research problem and indicate how it can be solved by the authors. I suggest full review.
The topic “Data Source” is not very explanatory, it does not demonstrate the types of data used, only showing where they were obtained. It lacks a more technical explanation of the data and complete citations of the places of acquisition.
The methods are completely scarce, explaining only a few techniques. I missed a more detailed explanation of how the work's methods can effectively contribute to solving the research problem. The method is the “heart” of the article and, in this case, the chapter is not consistent with the quality of sustainability.
The results are interesting and well explained, however the maps differ from the results. The quality of the maps is low, mainly due to the lack of basic cartographic elements, such as explanatory captions and geographic coordinates. I suggest reviewing all the maps produced for the article.
I believe the article is interesting and has potential for publication. However, I suggest restructuring the text's writing so that it fits the quality of Sustainability. Introduction, methods and conclusion must be rewritten and improved so that the article can be accepted.
Author Response
Please see the attached PDF file, thanks.

Reviewer 2 Report
The topic is interesting, unfortunately the work lacks the features of a scientific article, including:
Lack of a clearly formulated research goal,
There is no reliable study of international literature,
A diagnosis of the current state is presented without the aspect of modelling, variants or predictions.
Detailed notes:
L 169 - ? to remove
L 179 "between the total number of fitness facilities and the total number of fitness facilities" please explain clearly the difference between the total number of fitness facilities and the total number of fitness facilities
The caption under figure 1 needs to be corrected
maps 1, 2 and 4 require correction - the geometry has not been preserved none of the maps has all the required elements of cartographic presentation
2.2.3. Research on the relationship between site layout of public fitness facilities and large medical sites in the main urban area of Beijing - In my understanding, the study of the relationship is unjustified, the relationship of a close location of fitness centers - a hospital? To what extent do the services include rehabilitation? more detailed analysis is needed.
L319 - L321 A similar conclusion would probably accompany analyzes covering other large cities: Paris, London ...
Author Response

(The authors gave the same response as above.)

Reviewer 3 Report
Manuscript Number: sustainability-2076560
Title: Study on rationality of public fitness service facilities in Beijing based on GIS
The manuscript "Study on rationality of public fitness service facilities in Beijing based on GIS" offers a useful method for examining how well-integrated public fitness service facilities and schools (kindergartens and primary schools) are in Beijing's major urban area. The design of public fitness facilities is correlated positively with housing costs. This is a significant and fascinating subject. I appreciate all the authors' hard work. This manuscript is unique and well-discussed based on the gathered information. Minor corrections have been made to this manuscript, though, to raise its quality.
Reviewer’s Review Comments to Author:
1. Originality: Authors have used novel thought in their study, which is suitable for this journal.
2. Scientific Quality: The work with the title “Study on rationality of public fitness service facilities in Beijing based on GIS " is very well structured, scientific and is written in a way that is easy to understand.
3. Relevance to the Field(s) of this Journal: The article is relevant to this journal as the aim and scopes have been matched.
4. General Comment: I appreciate the hard work put in by the writers. This is an original work that does an excellent job of discussing the data that was acquired. The methodology is good but needs more robust information in the introduction, analysis, and discussion.
5. Abstract and Keywords:
a. Authors may avoid numbering in the abstract and rewrite.
b. At least 5 keywords should be given.
6. Introduction: The author should write a detailed description in this section.
7. Literature Review: Authors should work on this section and it is recommended to cite a few more recent works to enrich your study, if possible, otherwise it is alright.
8. Data source and study Methods:
a. Authors may add a table to explain “Data Source”.
b. Authors should add a location map in the “Study area”.
c. Entire methodology may be framed in a flowchart.
9. Results: Authors should add coordinates in the maps.
10. Discussions: On page 13, line no 396 authors may omit the term “insufficient” and replace it with “deficiencies”.
11. Conclusions: Authors may avoid numbering in this section and present it more vivid way.
12. References / Bibliography: All fine
13. Figures: It has already been mentioned in the methodology and result section.
14. Tables: Ok.
15. Others: Nothing More
16. Reviewer’s Decision Comment: Some revisions are essential before the final acceptance.
Best wishes.
Author Response

(The authors gave the same response as above.)

Round 2
Reviewer 1 Report
The paper is ready to be accepted
Author Response
Response to the comments
Dear Reviewer,
Thank you very much for your decision. All of your previous comments are all valuable and very helpful for revising and improving our manuscript.
We hope the revised manuscript could meet the requirements for publication in Sustainability.
Sincerely,
Xuefeng Tan
Reviewer 2 Report
Dear Authors,
Thank you for your detailed comments - I accept them.
Please refer to the general comments contained in the previous review, including:
Lack of a clearly produced research goal,
There is no reliable study of international literature,
A diagnosis of the current state is presented without the aspect of modeling, variants or predictions.
Regards
Author Response
Dear Reviewer,
Thank you very much for your decision and the comments. These comments are all valuable and very helpful for revising and improving our manuscript. We have revised the manuscript carefully and accordingly. The revisions have been marked in read in the revised manuscript. We have provided a list of our response to the reviewers' comments.
We hope the revised manuscript could meet the requirements for publication in Sustainability. We look forward to hearing from you soon.
Sincerely,
Xuefeng Tan
Question 1:Please refer to the general comments contained in the previous review, including:
- Lack of a clearly produced research goal,
- There is no reliable study of international literature,
- A diagnosis of the current state is presented without the aspect of modeling, variants or predictions.
Response: Thank you for your suggestions. Based on your first question and suggestion, I have made changes to clearly state the research objectives. The specific modifications are as follows:
In summary, this paper visualizes and analyzes the public fitness service facilities in Beijing based on the population agglomeration index and consistency test commonly used in population geography, ArcGIS 10.7 software. The main objectives are: (1) to analyze the characteristics of the spatial layout of public fitness service facilities in the main urban area of Beijing through hotspot analysis, coupling, and correlation coefficients; (2) to conduct consistency tests on the layout and number of public fitness service facilities in the main urban area of Beijing to determine whether the public fitness service facilities in the main urban area of Beijing are reasonable and the ratio of the observed distance to the reasonable expected distance of the near-neighbor index; (3) to provide feasibility advice on the development of public construction services in the main urban area.
Thank you for your suggestion. In response to your second suggestion, I have added relevant references, which include reliable international literature as well as recent studies. We hope these changes can address your concern.
Thank you for your advice. I'm very sorry to make your confused. I have removed the previous Section 1.2.5 model in order to make the full text more streamlined. This method is no longer used in this article.
The model mentioned in the previous Section 1.2.5 is to establish a coupling and coordinated development model between public fitness service facilities and population concentration. It is used to judge the development relationship between population and service facilities. However, since this paper prefers to use the method to test the status quo of public fitness service facilities in the main urban area of Beijing and put forward relevant suggestions, and the method proposed in Section 1.2.3 and Section 1.2.4 has been used for calculation, so previous section 1.2.5 is removed to make the whole paper complete. The purpose is clearer, readers and reviewers can better understand the content of the article.
The main objective of this paper is to use GIS technology to investigate the current situation and propose solutions. Your suggestion will be adopted in the follow-up research. Thank you again for your suggestion, and I am deeply sorry for confusing you.
Reviewer 3 Report
Manuscript id: sustainability-2076560
Title: Study on rationality of public fitness service facilities in Beijing based on GIS
Comments:
All of the reviewer's questions received in-depth answers from the author. They updated their work to reflect the reviewer's suggestions, making any necessary additions and clarifications. The reviewers' suggestions greatly improved the revised manuscript. Before publishing the manuscript, carefully check for mistakes. Examples include changing "arcgis10.7 software" to "ArcGis 10.7 software" in the abstract section of this manuscript and line 12.
Decision: The paper should be accepted for publishing in your prestigious journal, in my opinion.
Author Response
Response to the comments
Dear Reviewer,
Thank you very much for your decision and the comments. These comments are all valuable and very helpful for revising and improving our manuscript. We have revised the manuscript carefully and accordingly. The revisions have been marked in read in the revised manuscript. We have provided a list of our response to the reviewers' comments.
We hope the revised manuscript could meet the requirements for publication in Sustainability. We look forward to hearing from you soon.
Sincerely,
Xuefeng Tan
Question 1:All of the reviewer's questions received in-depth answers from the author. They updated their work to reflect the reviewer's suggestions, making any necessary additions and clarifications. The reviewers' suggestions greatly improved the revised manuscript. Before publishing the manuscript, carefully check for mistakes. Examples include changing "arcgis10.7 software" to "ArcGis 10.7 software" in the abstract section of this manuscript and line 12.
Response: Thank you for your suggestion. I have revised and checked the details of the article. All of your previous comments are all valuable and very helpful for revising and improving our manuscript. We have changed "arcgis10.7 software" to "ArcGis 10.7 software" accordingly. We hope the revised manuscript could meet the requirements for publication in Sustainability.
Round 3
Reviewer 2 Report
Accept in present form